# ROSA: An Optimization Algorithm for Multi-Modal Derivative-Free Functions in High Dimensions

**Ilija Ilievski**
College of Design and Engineering
National University of Singapore
ilija@u.nus.edu

**Wenyu Wang**
College of Design and Engineering
National University of Singapore
wenyu_wang@u.nus.edu

**Christine A. Shoemaker**
Civil and Environmental Engineering
Cornell University
cas12@cornell.edu

## Abstract

Derivative-free, multi-modal optimization problems in high dimensions are ubiquitous in science and engineering. Obtaining satisfactory solutions to high-dimensional optimization problems requires many objective function evaluations. At the same time, commonly used Bayesian optimization methods are typically computationally too expensive for sufficient sampling of the high-dimensional space which limits their function approximation accuracy and leads to sub-optimal solutions. We propose ROSA, a novel optimization algorithm based on well-known optimization techniques such as randomized optimization, simulated annealing, and surrogate optimization. ROSA is several orders of magnitude computationally more efficient than leading scalable Bayesian optimization methods, while also obtaining comparable or better solutions with as many as 4 times fewer objective function evaluations. We compare ROSA with a diverse set of methods on many synthetic high-dimensional benchmark functions and real-world problems.

## 1 Introduction

Many important problems in science and engineering require the optimization of multi-modal derivative-free functions, often resource-limited to only several thousand function evaluations. For example, calibration of water simulation models [1, 2], climate simulation models [3, 4], aircraft wing design [5–7], vehicle design optimization [8], and machine learning hyperparameter optimization [9–11]. With the proliferation of data and computing power, the functions of interest have become increasingly higher-dimensional. Thus, we focus on optimizing functions in 50 to 500 dimensions and with an evaluation budget limited at most ten times the dimension.

Applying optimization methods to high-dimensional functions is a difficult problem due to the curse of dimensionality, that is, the volume of the sampling space and the required function evaluations for an accurate function approximation grow exponentially with the number of dimensions. One line of research, led by REMBO, attempts to avoid the problem by embedding the high-dimensional space in a lower-dimensional space where the optimization is performed [12]. A closely related method is SASSBO which optimizes only some of the dimensions [13]. However, REMBO's assumption that the high-dimensional function can be accurately represented in low dimensions is often not valid in practice, and SASSBO's high computational requirements limit its application to problems with evaluation budgets of at most hundred function evaluations. Another line of research is the

Workshop on Bayesian Decision-making and Uncertainty, 38th Conference on Neural Information Processing Systems (NeurIPS 2024).

scalable Bayesian optimization methods, which aim to fit a sparse Gaussian process models [14], or fit a Gaussian process models only on subset of evaluated points around a trust region [11]. But, sparsifying the solutions space or using only subset of the evaluated points often results in suboptimal solutions. Finally, the inherit computational expense of fitting and tuning Gaussian process models, commonly limits the application of these models to problems with evaluation budgets of at most thousand iterations.

We propose a fundamentally different optimization algorithm that combines metaheuristic search method with a polyharmonic spline surrogate. Polyharmonic splines are fitted in closed-form and require no tuning, in addition to being accurate approximates of high-dimensional functions [15]. Our metaheuristic search method merges ideas from randomized optimization and simulated annealing to achieve optimal exploration-exploitation trade-off and efficiently avoid local optima. To summarize our contributions are: 1. We develop ROSA, a novel optimization algorithm for efficient multi-modal optimization of high-dimensional derivative-free functions with an evaluation budget of several thousand evaluations. 2. We open-source our modern and efficient implementation of ROSA, able to run on CPUs and GPUs. We hope our code will be used by researchers and practitioners in a wide range of applications in science and engineering.

## 2   Method

We develop an optimization method for a multi-modal derivative-free function over a hypercube defined by $\mathbf{a}$ and $\mathbf{b}$ in $d$ dimensions. That is $\mathbf{x}^* = \operatorname{argmin}_{\mathbf{x}^\dagger \in [\mathbf{a}, \mathbf{b}]^d} f(\mathbf{x}^\dagger)$, notation info in Sec. A.1.

The proposed algorithm, ROSA, is based on three fundamental ideas in multi-modal, derivative-free optimization: (i) *Randomized Optimization*, where one randomly samples points by adding a random vector to the current best point. However, instead of adding random perturbations to *all* dimensions, ROSA only changes a small and decreasing number of dimensions at each iteration. (ii) *Simulated Annealing*, where one allows accepting a worse neighbour as the current best point, but with a probability that decreases with the number of iterations. (iii) *Surrogate optimization*, where one fits a surrogate such as Gaussian Process and uses an acquisition function such as Expected Improvement to decide where to evaluate $f$ next. Instead, ROSA uses computationally efficient surrogate such as polyharmonic splines to only rank the neighbours and evaluate on $f$ the top-ranked neighbours.

The algorithm takes as inputs an objective function $f$ we wish to minimize and an evaluation budget $n_{\max}$. ROSA starts by evaluating the objective function at $n_0$ points, sampled uniformly at random from the function input space. The point that gives the lowest objective function value is set as the current best point, $\mathbf{x}'$, from where the optimization iterations start.

At each iteration, ROSA selects a set of dimensions to be perturbed at random, with a probability of being selected $p_\varphi = \varphi(n/n_{\max})$ . The probability of each dimension being selected is a decreasing function ($\varphi$) of the amount of the currently used evaluation budget (Sec. 2) and it is independent from the selection of the other dimensions. Formally, we define the set of selected dimensions: $\mathcal{A} = \{k : \upsilon < \varphi(n/n_{\max}) \mid k \in \mathcal{I}, \ \upsilon \sim U(0,1)\}$, where $\mathcal{I} = \{1, \ldots, d\}$ and $U(0,1)$ is Uniform distribution over $(0, 1) \in \mathcal{R}$. In case, $\mathcal{A} = \emptyset$, then $\mathcal{A} = \{j\}$ where $j$ is a random sample from $\mathcal{I}$, ensuring at least one dimension is always selected. The probability of a dimension being selected is also independent across iterations, resulting in different dimensions being selected at each iteration.

We create a set $\mathcal{C}$ of $q$ neighbouring points, with $q \gg d$, i.e. $\mathcal{C} = \{\hat{\mathbf{x}}_i \mid i \in \{1, \ldots, q\}\}$, by adding random perturbations to the selected dimensions of the current best point $\mathbf{x}'$ (Sec. 2). Selecting the dimensions is independent across the $\hat{\mathbf{x}}_i$ points, so each point may have different dimensions selected. We then evaluate each $\hat{\mathbf{x}}_i$ point in $\mathcal{C}$ on a polyharmonic spline surrogate and select the point with lowest surrogate value as the next evaluation point $\mathbf{x}^\circ$.

The point $\mathbf{x}^\circ$ is evaluated on the objective function and it is accepted as the current best point with probability $p_\alpha = g(f(\mathbf{x}^\circ), f(\mathbf{x}'), n)$, given by $g$, the acceptance probability function (Sec. 2). After exhausting the evaluation budget we return the current best point and its objective value. Note that ROSA is easily parallelizable by choosing the $m$ lowest surrogate value points instead of a single point, given that one is able to run the objective function efficiently in parallel.

Next, we describe the algorithm's components, and in the appendix, we define the algorithm in pseudo code (Alg. 1), provide a proof of convergence (Sec. A.2), and open-source our code at `https://github.com/ili3p/ROSA`.

**Algorithm 1** Randomized Optimization with Simulated Annealing (ROSA)
___
**Inputs:**

    1. Objective function $f(\mathbf{x}) : \mathbb{R}^d \to \mathbb{R}^1$, with $\mathbf{x} \in [\mathbf{a}, \mathbf{b}]^d \subset \mathbb{R}^d$, where $\mathbf{a}$ and $\mathbf{b}$ are $d$-dimensional vectors that delineate the objective function domain.

    2. Evaluation budget $n_{\max}$.

**Step 1:** Initialization

    (a) Sample uniformly at random $n_0$ points and define the set $\mathcal{D}_{n_0}$ as a set of pairs of point and its function value, $\mathcal{D}_{n_0} = \{(\mathbf{x}_1, f(\mathbf{x}_1)), \ldots, (\mathbf{x}_{n_0}, f(\mathbf{x}_{n_0})) \mid \mathbf{x}_i \sim U(\mathbf{a}, \mathbf{b})\}$.

    (b) Define $\mathbf{x}'$ as $(\mathbf{x}', f(\mathbf{x}')) \in \mathcal{D}_{n_0}$ such that $f(\mathbf{x}') \leq f(\mathbf{x}_i) \,\forall\, (\mathbf{x}_i, f(\mathbf{x}_i)) \in \mathcal{D}_{n_0}$.

    (c) Define $y'$ as the current lowest found objective value, $y' = f(\mathbf{x}')$.

    (d) Define $n$ as the number of currently evaluated points, $n = n_0$.

**Step 2:** Main optimization loop

`while` $n \leq n_{\max}$ `do`

    (a) Fit a polyharmonic spline surrogate $H$ on $\mathcal{D}_n$ (Equation 2).

    (b) Create a set $\mathcal{C}$ of $q$ neighbouring points around $\mathbf{x}'$, $\mathcal{C} = \texttt{NG}(\mathbf{x}', n/n_{\max})$ (Algorithm 2).

    (c) Define the set $\mathcal{S}$ as a set of pairs of neighbour point and its surrogate value, $\mathcal{S} = \{(\hat{\mathbf{x}}_i, H(\hat{\mathbf{x}}_i)) \mid \hat{\mathbf{x}}_i \in \mathcal{C}\}$.

    (d) Denote $\mathbf{x}^\circ$ as the best point in $\mathcal{S}$, i.e. $H(\mathbf{x}^\circ) \leq H(\hat{\mathbf{x}}_i) \,\forall\, (\hat{\mathbf{x}}_i, H(\hat{\mathbf{x}}_i)) \in \mathcal{S}$.

    (e) Perform a function evaluation $y^\circ = f(\mathbf{x}^\circ)$ and update $\mathcal{D}_{n+1} = \{(\mathbf{x}^\circ, y^\circ)\} \cup \mathcal{D}_n$.

    (f) With probability $p_\alpha = g(y^\circ, y', n)$ (Section 2) accept the new solution $\mathbf{x}^\circ$ and set $\mathbf{x}' = \mathbf{x}^\circ$ and $y' = y^\circ$.

    (g) $n = n + 1$.

`end`

**Return:** The lowest objective value and the corresponding point, $y'$ and $\mathbf{x}'$.
___

**Polyharmonic Spline Surrogate** A polyharmonic spline surrogate is a linear combination of radial basis functions and a polynomial tail. The polyharmonic kernels are scale invariant, so they do not require tuning of hyperparameters such as the length-scale in the case of the squared exponential and Matérn kernels, the most common Gaussian Process kernels. The polyharmonic spline surrogate is defined as: $\phi(r) = r^k$ if $k = 2n - 1$ for $n \in \mathbb{N}$ and as $\phi(r) = r^k \ln(r)$ if $k = 2n$ for $n \in \mathbb{N}$, where $r = \|\mathbf{x} - \mathbf{c}_i\|_2$ and $\mathbf{c}_i$ are the centres, i.e., the evaluated points used to fit the surrogate. ROSA incorporates a cubic spline, $\phi(r) = r^3$, which is conditionally positive definite kernel of order 2. So, we include a linear polynomial tail to ensure the stability of the solutions of the system of equations used to fit the surrogate parameters. Formally, given $n$ number of $d$-dimensional vectors, $\mathbf{x}_{1:n}$, we construct an polyharmonic spline interpolation model with:

$$H(\mathbf{x}) = \sum_{i=1}^{n} \lambda_i (\|\mathbf{x} - \mathbf{x}_i\|_2)^3 + \mathbf{b}^\top [1, \mathbf{x}^\top] \tag{1}$$

The model parameters $\lambda_{i:n}, b_{i:d}$ are determined by solving the following linear system of equations:

$$\begin{bmatrix} \boldsymbol{\Phi} + \eta & \boldsymbol{P} \\ \boldsymbol{P}^\top & \mathbf{0} \end{bmatrix} \begin{bmatrix} \boldsymbol{\lambda} \\ \boldsymbol{b} \end{bmatrix} = \begin{bmatrix} \boldsymbol{F} \\ \mathbf{0} \end{bmatrix} \tag{2}$$

Here $\boldsymbol{\Phi} \in \mathbb{R}^{n \times n}$ is defined as $\Phi_{i,j} = (\|\mathbf{x}_i - \mathbf{x}_j\|_2)^3, i, j = 1, \ldots, n$, $\mathbf{0} \in \mathbb{R}^{(d+1) \times (d+1)}$, $\mathbf{P} \in \mathbb{R}^{n \times (d+1)}$ has its $i$-th row defined as $[1, \mathbf{x}_i^\top]$, $\boldsymbol{\lambda} = [\lambda_1, \ldots, \lambda_n]^\top$, $\boldsymbol{F} = [f(\mathbf{x}_1), \ldots, f(\mathbf{x}_n)]^\top$, and $\eta$ is a regularization constant. When $n < d + 1$, we fit the model parameters via least squares.

**Acceptance Probability Function** The acceptance probability function promotes exploration by accepting with some probability a point with a worse objective function value as the current best point [16]. We define the acceptance function as $p_\alpha := g(f(\mathbf{x}^\circ), f(\mathbf{x}'), n) = \min(\exp[\frac{-(f(\mathbf{x}^\circ) - f(\mathbf{x}'))}{T_n}], 1)$, where $f(\mathbf{x}^\circ)$ is the new, possibly worse, objective function value, and $f(\mathbf{x}')$ is the current best objective value. $T_n$ is the cooling temperature with geometric schedule, i.e., $T_n = \alpha^n T_0$, where $T_0$ and $\alpha$ are set to common values [17] and fixed throughout our experiments. We accept $\mathbf{x}^\circ$ as the current best point with probability $p_\alpha$.

**Probability of Perturbing a Dimension**  We define the function $\varphi(n/n_{\max})$ as a simple decreasing step-function that goes from $1\mathrm{e}{-}1$ to $1\mathrm{e}{-}6$ (see code for the specific implementation). In Sec. B.5, we analyse our choice of probabilities for selecting dimensions by comparing the default ROSA, with ROSA-Const. 20D where we perturb $\min(20, d)$ dimensions, ROSA-Const. 10% where we use constant function $\varphi(\cdot) = 1\mathrm{e}{-}1$, ROSA-Increasing, where we *increase* the probabilities from $0.10$ to $1.00$, ROSA-Reversed Default, where we increase the probabilities from $1\mathrm{e}{-}6$ to $1\mathrm{e}{-}1$. In Figure B.3 we observe that the performance of ROSA is similar as long as the probabilities are decreasing.

**Perturbation Distribution**  To the selected dimensions of the current best point, we add random samples from truncated normal distribution (Eq. 5) with mean 0 and perturbation radius fixed to one sixth of the objective function domain range, i.e., $\sigma = (\mathbf{b} - \mathbf{a})/6$. It is important that the samples are coming from truncated normal distribution, instead for example from uniform distribution [11], as otherwise the distribution of the samples gets heavily skewed towards the domain bounds (Figure B.2). We recommend that $\sigma$ is set to one sixth of the bounded range such that $99.7\%$ of the hypercube space is reachable from the centre point. To empirically justify our choice, we benchmark ROSA with multiple $\sigma$ values and show that ROSA is fairly robust to different values of $\sigma$, with the default version only having a small advantage on some functions (Figure B.4). The pseudo code is in Alg 2.

---

**Algorithm 2** Neighbour Generation in ROSA

---

**Inputs:**
1. Point $\mathbf{x}'$ around which to generate neighbouring points.
2. Percentage of currently spent evaluation budget $\eta = n/n_{\max}$.

**Configuration:**
1. Bounding hypercube $[\mathbf{a}, \mathbf{b}]^d \subset \mathbb{R}^d$ of permitted values of $\mathbf{x}$.
2. A *decreasing* step-function that maps a percentage of spent budget to a probability of perturbing a dimension, $p_\varphi = \varphi(\eta)$ (Section 2).
3. Number of neighbour points $q$.

**Algorithm:** Create a set $\mathcal{C}$ of $q$ neighbour points $\hat{\mathbf{x}}_i$ where for each point:
1. Define a set of selected dimensions: $\mathcal{A} = \{k : \upsilon < \varphi(\eta) \mid k \in \mathcal{I}, \ \upsilon \sim U(0,1)\}$, where $\mathcal{I} = \{1, \ldots, d\}$ and $U(0,1)$ is Uniform distribution over $(0,1) \in \mathcal{R}$.
2. Construct the permutation vector $\mathbf{z} = [z^{(k)}]^\top$, for $k \in \mathcal{I}$, $z^{(k)} = \tau \sim \mathcal{T}(0, \sigma, \alpha^{(k)}, \beta^{(k)})$, if $k \in \mathcal{A}$ otherwise, $z^{(k)} = 0$. Here $\mathcal{T}(0, \sigma, \alpha^{(k)}, \beta^{(k)})$ is truncated Normal distribution with mean 0 and standard deviation $\sigma$ bounded by $\boldsymbol{\alpha} = \mathbf{a} - \mathbf{x}'$ and $\boldsymbol{\beta} = \mathbf{b} - \mathbf{x}'$.
3. Set $\hat{\mathbf{x}}_i = \mathbf{x}' + \mathbf{z}$.

**Return:** The set of $\mathcal{C} = \{\hat{\mathbf{x}}_i \mid i \in \{1, \ldots, q\}\}$ neighbour points.

---

## 3  Experiments

ROSA combines ideas from simulated annealing, randomized optimization, and surrogate optimization. Accordingly we compare ROSA with baselines that employ ideas from simulated annealing, Dual Annealing [18], randomized optimization, PSO [19], and surrogate optimization (lq-CMA-ES [20], TuRBO [11], DYCORS [21]). We also compare ROSA to an evolutionary algorithm (CMA-ES) as it have been shown to be competitive in optimization of black-box functions with large evaluation budgets [22]. ROSA is most similar to TuRBO and DYCORS, the three methods add random perturbations to the current best, so we list their similarities and differences in Table B.1.

We compare the methods on three well-known synthetic functions, Ackley [23] — a function most surrogates can approximate well, Michalewicz [24] — a function with flat surface and sharp ridges where surrogates are often not helpful, and Rastrigin [25] — function riddled with many good local minima and thus algorithms often get stuck in a local minimum. 2D visualizations of the three functions are shown in Figure B.1. We vary the number of dimensions from 60 to 200 to evaluate how the algorithms' performance scale with the number of dimensions. As an additional challenging benchmark suit we use the BBOB-largescale suite from COCO [26].

As real-world problems we use the well-known benchmark problem of optimizing vehicle design introduced by General Motors at MOPTA08 [8] (in 124D), and optimizing 496 portfolio weights

for maximizing volatility adjusted returns while minimizing max drawdown, a common finance application of non-convex optimization methods. We perform parallel optimization with $m = 50$ on problems with a budget of more than 2,000 evaluations, such as BBOB-320D, BBOB-640D, and Portfolio Optimization, on the rest we perform serial optimization ($m = 1$). For details see Sec. B.1.

| | Ackley | | | | Michalewicz | | | | Rastrigin | | | | MOPTA | PRTF |
|---|---|---|---|---|---|---|---|---|---|---|---|---|---|---|
| Method \Dim. | 60 | 120 | 150 | 200 | 60 | 120 | 150 | 200 | 60 | 120 | 150 | 200 | 124 | 496 |
| Sobol | 12.39 | 13.01 | 13.15 | 13.28 | -13 | -23 | -27 | -34 | 869 | 1854 | 2349 | 3200 | 308 | - |
| PSO | 13.35 | 13.52 | 13.67 | 13.60 | -17 | -28 | -33 | -42 | 898 | 1897 | 2413 | 3218 | 316 | -36.9 |
| DA | 12.69 | 13.10 | 13.17 | 13.34 | -12 | -21 | -26 | -32 | 878 | 1856 | 2349 | 3213 | 313 | - |
| CMA-ES | 8.74 | 8.65 | 8.47 | 8.50 | -13 | -23 | -27 | -35 | 714 | 1533 | 1953 | 2697 | 251 | -53.7 |
| lq-CMA-ES | 4.14 | 4.46 | 4.58 | 4.63 | -13 | -22 | -27 | -33 | 669 | 1293 | 1620 | 2168 | 235 | - |
| TuRBO | 4.56 | 5.77 | 6.20 | 6.38 | -24 | -54 | -69 | -92 | 421 | 805 | 1008 | 1334 | 243 | -67.0 |
| DYCORS | **2.23** | 2.51 | 2.53 | 2.6 | -32 | -68 | -86 | -113 | 286 | 569 | 711 | 957 | - | - |
| ROSA | 2.69 | **1.99** | **1.90** | **1.8** | **-35** | **-71** | **-89** | **-118** | **272** | **520** | **653** | **884** | **226** | **-77.9** |

Table 1: Mean best value obtained from 30 independent trials, across problems and dimensions, for each method under comparison, with bold we denote the best overall result. PSO - particle swarm optimization, DA - Dual Annealing.

**Results** The results in Table 1 show that ROSA significantly outperforms all methods under comparison on all functions across dimensions (with the exception of DYCORS on Ackley 60D). Further, ROSA advantage over the other methods increases with increasing number of dimensions, showing that ROSA is especially suited for the high-dimensional optimization problems that are becoming increasingly ubiquitous. Given the no free lunch theorem, we do not claim ROSA will outperform any method on any function. However, on multi-modal derivative-free functions, such as the representative set of functions in our benchmark, ROSA is expected to outperform.

The progress plots in Sec. C show that not only ROSA achieves lowest objective value after exhausting the evaluation budget, but it also consistently outperforms most methods on most problems after any number of iterations. For example, on Ackley 200D (Figure C.2 right), ROSA achieves the lowest value found by TuRBO, the scalable Bayesian optimization method, after only using $27\%$ of the evaluation budget. Which means ROSA achieved comparable solution to TuRBO with almost 4 times fewer function evaluations. Furthermore, ROSA's code is two orders of magnitude faster than TuRBO's (Figure C.10). This means, as opposed to scalable Bayesian optimization methods, ROSA can be applied to difficult problems necessitating otherwise prohibitively large number of evaluations.

The outperformance of ROSA is also confirmed on the ten multimodal functions in high dimensions of the BBOB benchmark suite. In 160D ROSA performance is matched by DYCORS after $75\%$ of the budget (Figure C.8). However, DYCORS computational requirements are too great to be run on the 320D and 640D problems (Figure C.9). Which once again confirms the need of a method with not only good optimization performance but also that is scalable and computationally efficient to be able to optimize high-dimensional problems and with many function iterations.

We perform sensitivity analysis of ROSA's crucial algorithm component, the neighbouring points generation. We justify our choice of probability of perturbing a dimension with the results in Figure B.3, the way we set the standard deviation of the perturbation distribution in Figure B.4, and our choice of perturbation distribution in Figure B.5.

## 4 Discussion

With this paper, we aim to address the problem of optimization of multi-modal, derivative-free functions in high dimensions by developing an optimization method that is computationally efficient, while achieving excellent optimization performance on 15 representative functions in 9 distinct dimensions. We hope ROSA will serve as an alternative to Bayesian optimization methods when they are not suitable for the task. ROSA's modular algorithm design is based on fundamental optimization ideas developed and time-tested over decades of research. The modular design also allows for easy customization to specific problem types and further development of the algorithm. As an example, future work involves adapting the exploration vs exploitation trade-off to unseen problems by dynamically adjusting the probability of perturbing a dimension and the acceptance probability.

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

# A  Appendix

## A.1  Notation

| Notation | Description |
| --- | --- |
| $f$ | objective function we wish to minimize |
| $\mathbf{x}$ | a vector, objective function input, or a point, used interchangeably |
| $x_i^{(k)}$ | the $k$-th element, i.e., dimension, of the $i$-th vector $x$ |
| $d$ | number of objective function dimensions, i.e., input vector length $\mathbf{x}$ |
| $\mathbf{a}, \mathbf{b}$ | vectors, delineating the bounds of the objective function domain |
| $\hat{\mathbf{x}}$ | a neighbour vector, only evaluated on the surrogate |
| $\mathbf{x}'$ | the current best vector, i.e., an *evaluated* point with lowest objective function value |
| $\mathbf{x}^\circ$ | the current best neighbour, i.e., a point with lowest *surrogate* function value |
| $y'$ | current lowest objective function value found |
| $y^\circ$ | objective *function* value of the current best neighbour |
| $H(\hat{\mathbf{x}})$ | a surrogate mapping neighbour vectors $\hat{\mathbf{x}}$ to objective function value *estimates* |
| $\phi(\|\|\mathbf{x} - \mathbf{x}_i\|\|_2)$ | kernel function centred on the evaluated point $\mathbf{x}_i$ |
| $n$ | number of spent function evaluations |
| $n_{\max}$ | number of maximum function evaluations, i.e., the evaluation budget size |
| $n_0$ | number of function evaluations used for initialization |
| $\eta$ | percentage of evaluation budget spent, i.e., $n/n_{\max}$ |
| $p_\alpha$ | probability of accepting the solution $y^\circ$ as the current best |
| $g(y^\circ, y', \eta)$ | function outputting the probability $p_\alpha$ of accepting the solution $y^\circ$ |
| $p_\varphi$ | probability of selecting a dimensions for perturbation |
| $\varphi(\eta)$ | function outputting the probability $p_\varphi$ of selecting a dimension |
| $\mathcal{I}$ | the set of dimensions, i.e. $\mathcal{I} = \{1, \ldots, d\}$ |
| $\mathcal{A}$ | the set of selected dimensions for perturbation during neighbour generation |
| $\mathcal{C}$ | the set of neighbour points |
| $\mathcal{D}$ | the set of pairs of point and objective function value, used to fit the surrogate |
| $\xi$ | the set of evaluated points, i.e., the algorithmic running history |
| $\check{\mathcal{S}}$ | the set of pairs of neighbour point and surrogate value |
| $U(a, b)$ | Uniform distribution between $a$ and $b$ |
| $\mathcal{T}(0, \sigma, \mathbf{a}, \mathbf{b})$ | truncated Normal with mean 0 and standard deviation $\sigma$ bounded by $\mathbf{a}$ and $\mathbf{b}$ |

## A.2 Convergence Analysis

After $n \geq n_0$ times of objective evaluation, consider the current best $d$-dimensional neighbour vector $\mathbf{x}_n^\circ$ as a random vector. Define $\xi_n = \{\mathbf{x}_1^\circ, \ldots, \mathbf{x}_n^\circ\}$ to store all the evaluated points, i.e., the entire algorithmic running history after $n$ iterations.

Finally, let $\mathcal{Q}(\mathbf{x}, \delta)$ denote the open ball centred at $\mathbf{x}$ with radius $\delta$, i.e., $\mathcal{Q}(\mathbf{x}, \delta) = \{\mathbf{x}^\dagger \mid ||\mathbf{x} - \mathbf{x}^\dagger||_2 < \delta\}$, and $\Sigma(\xi_n)$ denote the $\sigma$-field generated by the random vectors in $\xi_n$.

**Theorem 1** *Let $f$ be a real-valued function defined on $\mathcal{B} \subseteq \mathbb{R}^d$ and suppose that $\mathbf{x}^*$ is the unique global minimizer of $f$ on $\mathcal{B}$ in the sense that $f(\mathbf{x}^*) = \inf_{\mathbf{x} \in \mathcal{B}} f(\mathbf{x}) > -\infty$. Following the pseudo code in Algorithm 1, ROSA iteratively generates random vectors $\{\mathbf{x}_n\}_{n \geq 1}$, and maintains a sequence of random vectors $[\mathbf{x}_n']$ as follows: $\mathbf{x}_n' = \mathbf{x}_n^\circ$ with probability $p_\alpha = g(f(\mathbf{x}^\circ), f(\mathbf{x}_n'), n)$ (Section 2) otherwise $\mathbf{x}_n' = \mathbf{x}_{n-1}'$. Then $\mathbf{x}_n'$ converges to $\mathbf{x}^*$ almost surely.*

*Proof.* Fix $\varepsilon > 0$ and $n \geq n_0 + 1$. Assume that there exists $\delta(\varepsilon) > 0$ such that $|f(\mathbf{x}) - f(\mathbf{x}^*)| < \varepsilon$ whenever $||\mathbf{x} - \mathbf{x}^*||_2 < \delta(\varepsilon)$. Hence, the event $[\mathbf{x}_n \in \mathcal{B} : |f(\mathbf{x}_n) - f(\mathbf{x}^*)| < \varepsilon] \supseteq [\mathbf{x}_n \in \mathcal{B} : ||\mathbf{x}_n - \mathbf{x}^*||_2 < \delta(\varepsilon)]$, and so,

$$
\begin{aligned}
P[\mathbf{x}_n \in \mathcal{B} : |f(\mathbf{x}_n) - f(\mathbf{x}^*)| < \varepsilon \mid \Sigma(\xi_{n-1})] &\geq P[\mathbf{x}_n \in \mathcal{B} : ||\mathbf{x}_n - \mathbf{x}^*||_2 < \delta(\varepsilon) \mid \Sigma(\xi_{n-1})] \\
&= P[\mathbf{x}_n \in \mathcal{Q}(\mathbf{x}^*, \delta(\varepsilon)) \cap \mathcal{B} \mid \Sigma(\xi_{n-1})] \\
&= \int_{\mathcal{Q}(\mathbf{x}^*, \delta(\varepsilon)) \cap \mathcal{B}} P(\mathbf{x}_n = \mathbf{x} \mid \Sigma(\xi_{n-1})) d\mathbf{x}
\end{aligned}
\tag{3}
$$

In ROSA, the neighbour $\mathbf{x}_n^\circ$ is generated by adding independent truncated normal samples to the selected dimensions of $\mathbf{x}_{n-1}'$. Let $\mathcal{E}_{n,j}$ denotes the event that the $j$-th dimension of $\mathbf{x}_{n-1}'$ is selected for perturbation and $P(\mathcal{E}_{n,j}) = p_{\varphi_n} = \varphi(n/n_{\max}) > 0$ (see Section. 2). Therefore, the candidate $\mathbf{x}_n^\circ$ has the following conditional density function given $\Sigma(\xi_{n-1})$ for each $n > n_0$,

$$
\begin{aligned}
P(\mathbf{x}_n^\circ = \mathbf{x} \mid \Sigma(\xi_{n-1})) &= \prod_{j=1}^d \quad P(x_{n,j}^\circ = x_j \mid \Sigma(\xi_{n-1})) \\
&= \prod_{j=1}^d \quad [P(x_{n,j}^\circ = x_j \mid \Sigma(\xi_{n-1}), \mathcal{E}_{n,j}) \cdot p_{\varphi_n} \\
&\qquad\quad + P(x_{n,j}^\circ = x_j \mid \Sigma(\xi_{n-1}), \neg\mathcal{E}_{n,j}) \cdot (1 - p_{\varphi_n})] \\
&\geq \prod_{j=1}^d \quad \mathcal{T}(x_{n,j}^\circ = x_j \mid \Sigma(\xi_{n-1})) \cdot p_{\varphi_n},
\end{aligned}
\tag{4}
$$

where $\mathcal{T}$ is a truncated normal density function in the following form,

$$
\mathcal{T}(x_{n,j}^\circ = x_j \mid \Sigma(\xi_{n-1}); \sigma_j, a_j, b_j) = \frac{1}{\sqrt{2\pi\sigma_j^2}} \frac{\exp\left[\frac{-(x_{n,j}^\circ - x_{n-1,j}')^2}{2\sigma_j^2}\right]}{\Phi(\frac{b_j - x_{n-1,j}'}{\sigma_j}) - \Phi(\frac{a_j - x_{n-1,j}'}{\sigma_j})}
\tag{5}
$$

with $\Phi$ being the cumulative function of a standard normal distribution and the rest defined in Section A.1. Bounding (5) from bellow:

$$
\mathcal{T}(x_{n,j}^\circ = x_j \mid \Sigma(\xi_{n-1}); \sigma_j, a_j, b_j) \geq \frac{1}{\sqrt{2\pi\sigma_j^2}} \frac{\exp\left[\frac{-(b_j - a_j)^2}{2\sigma_j^2}\right]}{\Phi(\frac{b_j - a_j}{\sigma_j}) - \Phi(\frac{a_j - b_j}{\sigma_j})} =: C_j > 0
\tag{6}
$$

Then, (4) is also bounded with:

$$
P(\mathbf{x}_n^\circ = \mathbf{x} \mid \Sigma(\xi_{n-1})) \geq \prod_{j=1}^d C_j \cdot p_{\varphi_n}.
\tag{7}
$$

Finally, combining (7) with (3) yields,

$$
\begin{aligned}
P[\mathbf{x}_n^\circ \in \mathcal{B} : |f(\mathbf{x}_n^\circ) - f(\mathbf{x}^*)| < \varepsilon \mid \Sigma(\xi_{n-1})] &\geq \int_{\mathcal{Q}(\mathbf{x}^*, \delta(\varepsilon)) \cap \mathcal{B}} \prod_{j=1}^d C_j \cdot p_{\varphi_n} d\mathbf{x} \\
&\geq \psi(\mathcal{Q}(\mathbf{x}^*, \delta(\varepsilon)) \cap \mathcal{B}) \prod_{j=1}^d C_j \cdot p_{\varphi_n} \\
&=: L(\varepsilon) > 0,
\end{aligned}
\tag{8}
$$

where $\psi$ is the Lebesgue measure on $\mathbb{R}^d$ and $p_{\varphi_n}$ denotes the probability used to select dimensions. Thus, from (8), we have $P[\mathbf{x}_n^\circ \in \mathcal{B} : |f(\mathbf{x}_n^\circ) - f(\mathbf{x}^*)| < \varepsilon \mid \sigma(\xi_{n-1})]$ is lower bounded by $L(\varepsilon)$. By following the same argument as in the proof of theorem in p.40 of [27], we show that $\mathbf{x}_n'$ converges to $\mathbf{x}^*$ almost surely. ∎

# B  Experimental Setup

## B.1  Methods

We compare ROSA with a diverse kind of methods, namely:

- Sobol, a quasi-random baseline represented by a scrambled Sobol sequence [28]; We use PyTorch's implementation of scrambled Sobol, available at link.

- PSO, a particle swarm optimization method [19]; We follow the instructions at link to install and run `pyswarms` on optimizing a function with bounds. We set options $= \{c1 : 0.5, c2 : 0.3, w : 0.9\}$, n_particles $= 10$, and use the `"GlobalBestPSO"` as optimizer with the rest settings set to default.

- DA, Dual Annealing, a method that combines classical simulated annealing with fast simulated annealing, coupled with a local search strategy [18]. We use the default values as implemented in SciPy [29].

- CMA-ES, a model-free evolutionary optimization method [22]; We follow the instructions at `https://github.com/CMA-ES/pycma` to install CMA-ES and run CMA with the default settings. As an initial point $\mathbf{x}_0$ we set the middle point of the domain, *i.e.* $\mathbf{x}_0 = (\mathbf{a} + \mathbf{b})/2$, where $\mathbf{a}$ and $\mathbf{b}$ are the upper and lower bounds respectively. Following CMA-ES authors advice, we set $\sigma_0$ to quarter of the objective function domain range.

- lq-CMA-ES, a surrogate-assisted evolutionary optimization method [20]; We follow the instructions at `http://cma.gforge.inria.fr/cmaes_sourcecode_page.html#practical` to install and run lq-CMA-ES with the default settings. As an initial point $\mathbf{x}_0$ we tried setting the middle point of the domain and using a random point from $n_0$ Latin Hypercube samples, the random LHS sample version worked better so we did all experiments with $\mathbf{x}_0$ set to a random LHS point. Following lq-CMA-ES authors advice on the above-mentioned URL, we set $\sigma_0$ to half of the function domain range, to make sure every space is within $2\sigma_0$ from every possible $\mathbf{x}_0$.

- TuRBO, a scalable Bayesian optimization method for high-dimensional surrogate optimization [11]; We follow the instructions at `https://github.com/uber-research/TuRBO/` to install TuRBO-1 and run TuRBO-1 with the default settings. We run only TuRBO-1 as our initial experiments with running TuRBO-m did not show any improvements over TuRBO-1 but significantly increased the required computational resources and wall-clock run time. Our initial experimental results are also supported by most experiments in [11].

- DYCORS, surrogate optimization method with dynamic coordinate search [21]. We use the implementation at `https://github.com/dme65/pySOT` to install pySOT and run DYCORS with the default settings [30, 31]. Due to pySOT's reliance on older software versions, we were not able to run DYCORS on MOPTA. DYCORS were also not run on the 496D portfolio optimization problem and 320D & 640D BBOB problems as it required unfeasible large amount of RAM memory.

ROSA is most similar to TuRBO and DYCORS, the three methods add random perturbations to the current best, so we list their similarities and differences in Table B.1.

## B.2  Benchmark functions

- Ackley is implemented as in `https://www.sfu.ca/~ssurjano/ackley.html` using the recommended values. We evaluate Ackley in the domain $\mathbf{x} \in [-5, 10]^D$ where $D \in \{60, 120, 150, 200\}$.

- Michalewicz is implemented as in `https://www.sfu.ca/~ssurjano/michal.html`. We evaluate Michalewicz in the domain $\mathbf{x} \in [0, \pi]^D$ where $D \in \{60, 120, 150, 200\}$.

| Probability of selecting a dimension | |
| --- | --- |
| DYCORS | $\min(20/d, 1) \cdot (1 - \ln(n)/\ln(n_{\max}))$ |
| TuRBO | fixed to $\min(20/d, 1)$ |
| ROSA | fixed to $[1e-1, 5e-2, 5e-3, 1e-6]$ |

| Perturbation distribution | |
| --- | --- |
| DYCORS | Mirrored Normal |
| TuRBO | Uniform |
| ROSA | Truncated Normal |

| Perturbation radius | |
| --- | --- |
| DYCORS | dynamically adjusted based on performance |
| TuRBO | dynamically adjusted based on performance |
| ROSA | fixed to $1/6$ of the domain range |

| Type of surrogate | |
| --- | --- |
| DYCORS | Cubic RBF with linear tail |
| TuRBO | Gaussian Process with Matérn 5/2 kernel |
| ROSA | Cubic RBF with linear tail |

| Exploration vs Exploitation trade-off | |
| --- | --- |
| DYCORS | Scoring function with cycling weights based on distance and surrogate value |
| TuRBO | Fixed on surrogate value |
| ROSA | Simulated annealing based acceptance function |

Table B.1: Similarities and differences of key algorithm components of ROSA, TuRBO, and DYCORS. The three methods are similar as they all add random perturbations to some dimensions of the current best point.

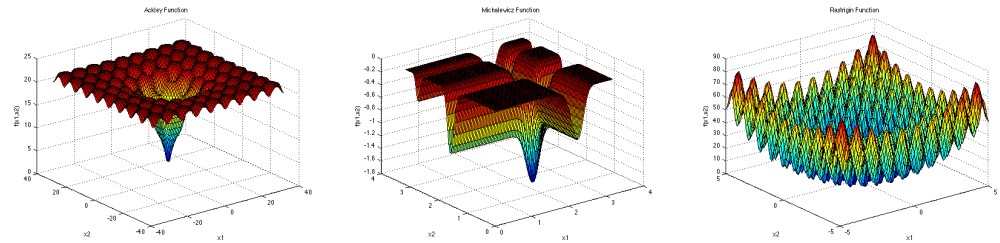

Figure B.1: 2D plots of the used benchmark functions. Ackley left, Michalewicz centre, and Rastrigin right.

- Rastrigin is implemented as in `https://www.sfu.ca/~ssurjano/rastr.html`. We evaluate Rastrigin in the domain $\mathbf{x} \in [-5.12, 5.12]^D$ where $D \in \{60, 120, 150, 200\}$.
- For BBOB we use the COCO suite [26] as implementation. We use the multi-modal functions, F15 through F24, on the following instances $[6, 12, 13, 9, 10, 12, 6, 1, 1, 2]$, chosen at random, and optimize in 160, 320, and 640 dimensions.

A two-dimensional plots of each of the functions are shown in Figure B.1.

## B.3 Real-world problems

We follow the instructions at `https://www.miguelanjos.com/jones-benchmark` to implement the MOPTA08 problem. As MOPTA08 is a constrained optimization problem, we transform it to unconstrained problem with $f_{\text{unconstr}}(\mathbf{x}) = f_{\text{constr}}(\mathbf{x}) + 10 \sum_{i=1}^{68} \max(0, c_i(\mathbf{x}))$ where $c_i \leq 0$ is a constrain violation. As usual, we optimize the function in $\mathbf{x} \in [0, 1]^{124}$.

The Portfolio Optimization problem aims to maximize volatility adjusted returns and minimize the maximum drawdown over the three year period of 2019-2022, of a portfolio consistent of 496 stocks that were part of SP 500 during all three years. Specifically, we minimize $m - r/(\sigma\sqrt{252})$, where $m$ is the portfolio maximum drawdown, $r$ is the total portfolio return after the three years, and $\sigma$ is the standard deviation of the daily returns.

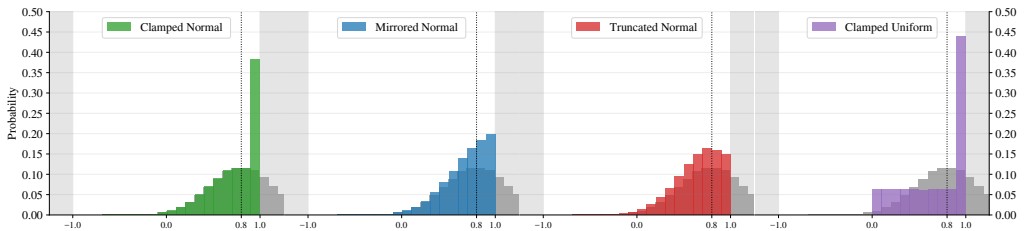

Figure B.2: Histograms of samples from different perturbation probability distributions for one dimensional point $x$ at $0.8$. Only samples from truncated normal distribution are not skewed towards the domain bound at $1.0$ as samples from the other distributions that fall outside the domain are set to the nearest bound.

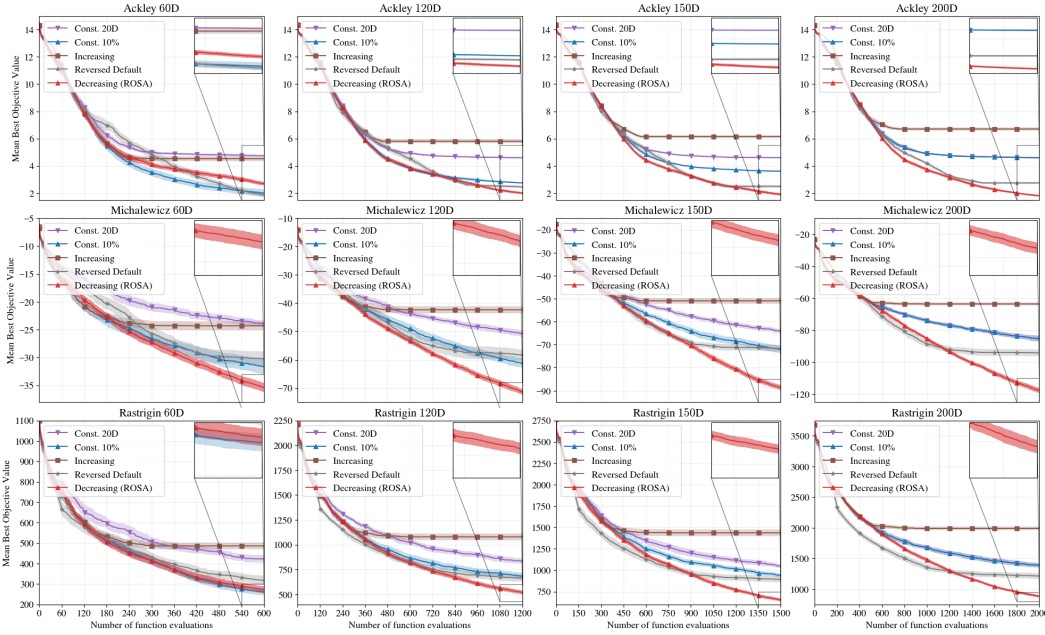

Figure B.3: **Method analysis**: Varying the probability function of selecting a dimension to be perturbed. Comparing the default ROSA, with ROSA-`Const. 20D` where we perturb $\min(20, d)$ dimensions, ROSA-`Const. 10%` where we use constant function $\varphi(\cdot) = 1e{-}1$, ROSA-`Increasing`, where we *increase* the probabilities from $0.10$ to $1.00$, ROSA-`Reversed Default`, where we increase the probabilities from $1e{-}6$ to $1e{-}1$.

## B.4 Computational resources

The experiments were run on a small cluster consisting of nodes with E5-2690v3 CPU and 32GB DDR4 RAM. Each experiment was limited to 24 hours of total wall-clock run time.

## B.5 Neighbour generation sensitivity analysis

# C Benchmark results

In this section we share the progress plots of optimizing the benchmark functions. Each line represents the mean of the lowest found objective value over 30 independent runs, with $95\%$ confidence bounds around it.

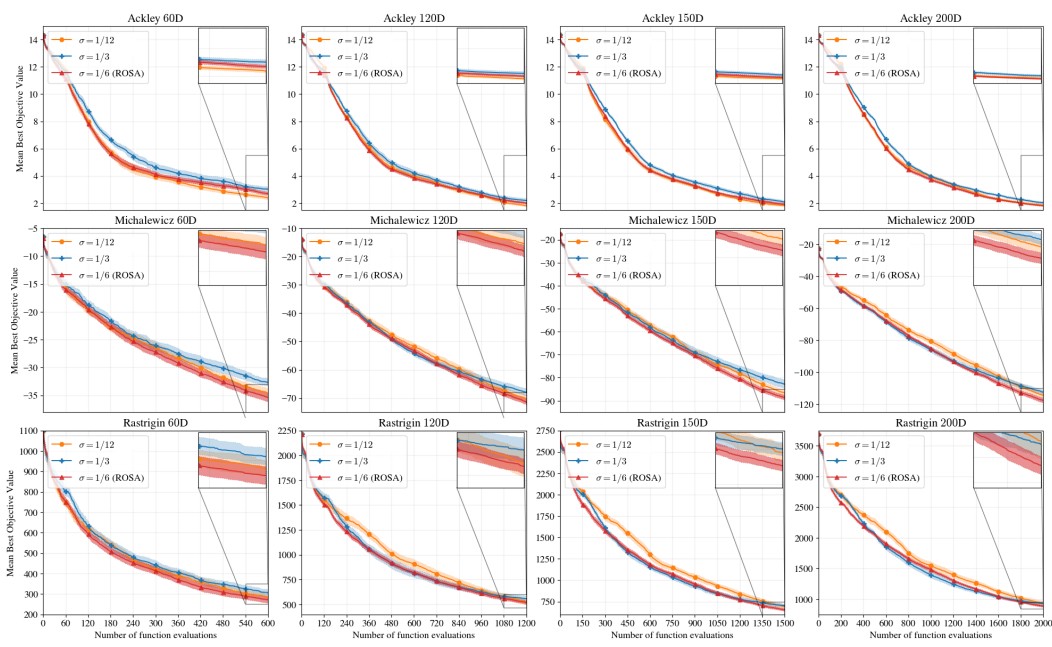

Figure B.4: **Method analysis**: Varying the standard deviation $\sigma$ of the truncated normal distribution used to generate perturbation samples.

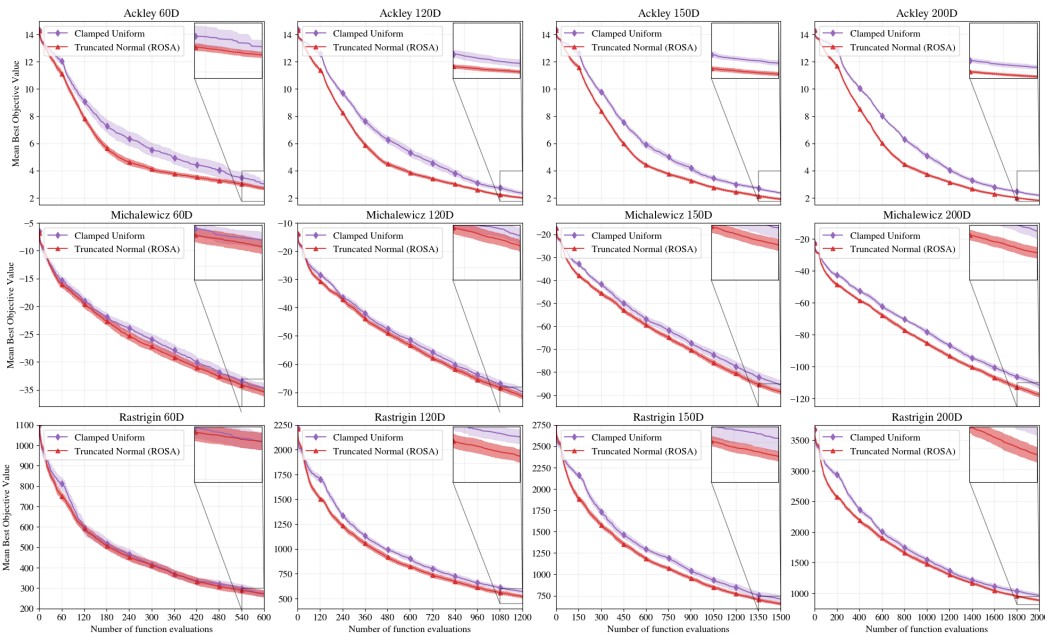

Figure B.5: **Method analysis**: Varying the distribution from which we sample perturbation samples. Comparing usning Clamped Uniform with the default — Truncated Normal.

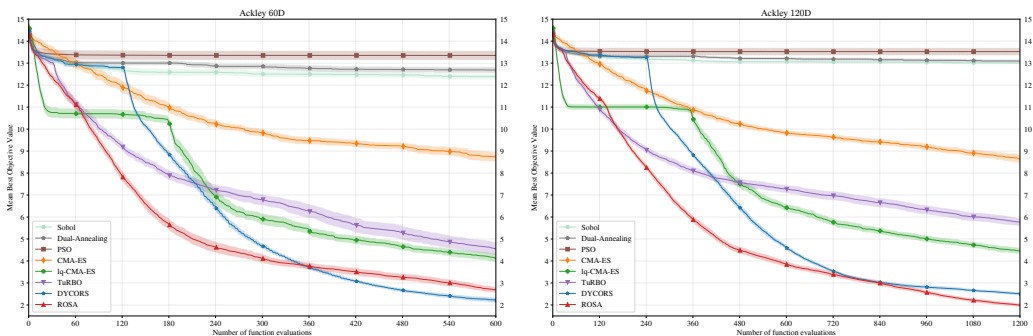

Figure C.1: Ackley optimization in 60 (left) and 120 (right) dimensions.

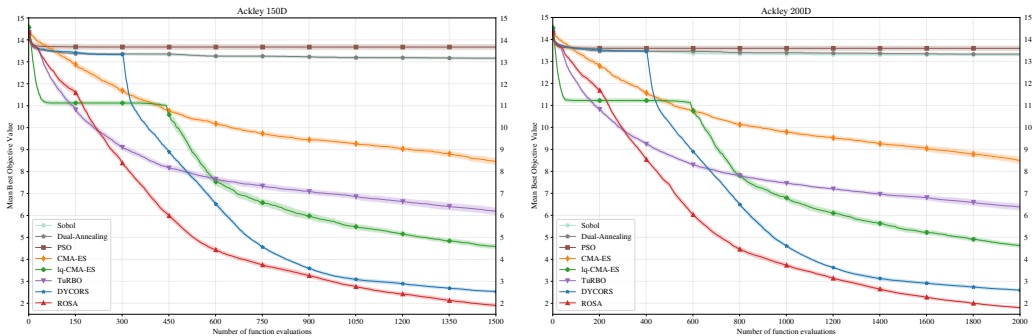

Figure C.2: Ackley optimization in 150 (left) and 200 (right) dimensions.

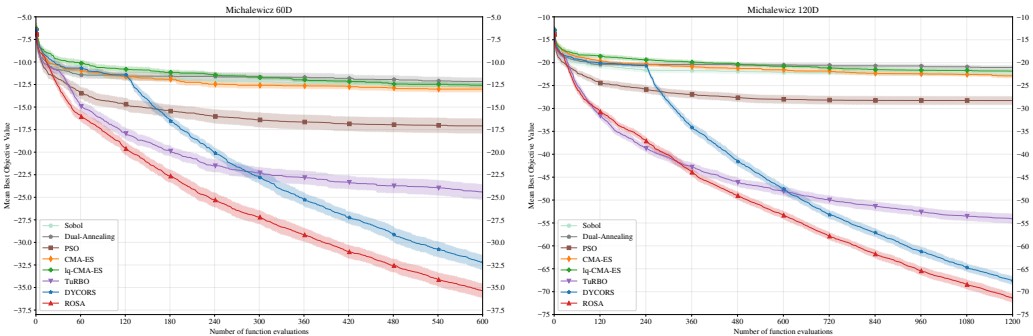

Figure C.3: Michalewicz optimization in 60 (left) and 120 (right) dimensions.

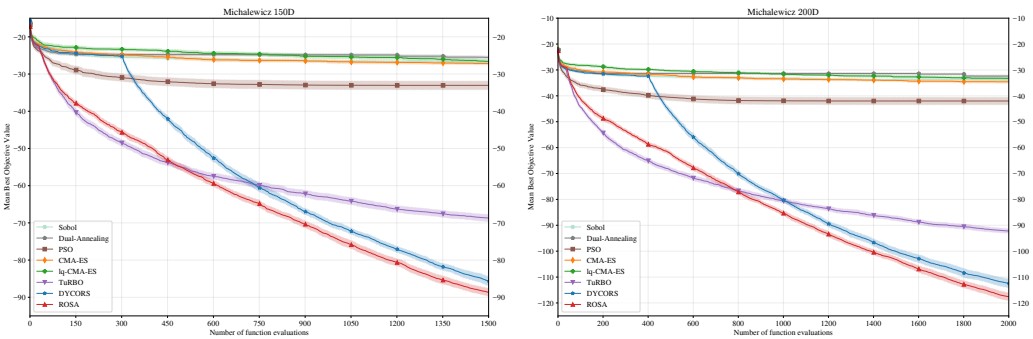

Figure C.4: Michalewicz optimization in 150 (left) and 200 (right) dimensions.

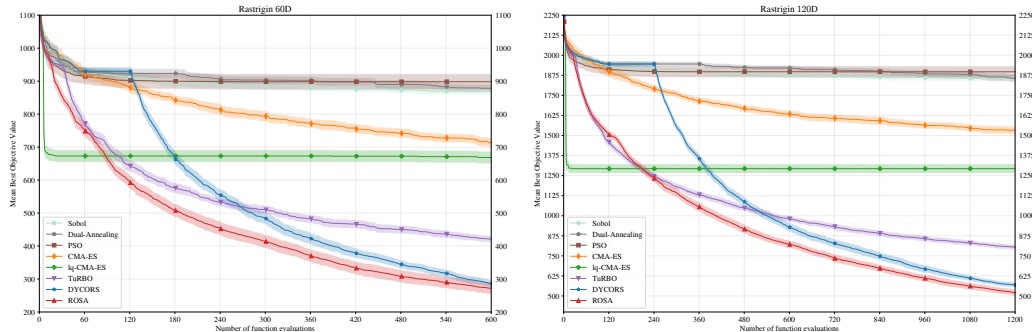

Figure C.5: Rastrigin optimization in 60 (left) and 120 (right) dimensions.

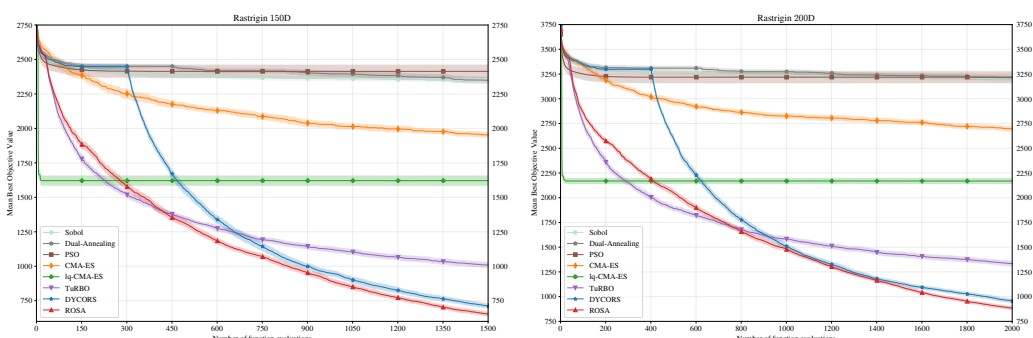

Figure C.6: Rastrigin optimization in 150 (left) and 200 (right) dimensions.

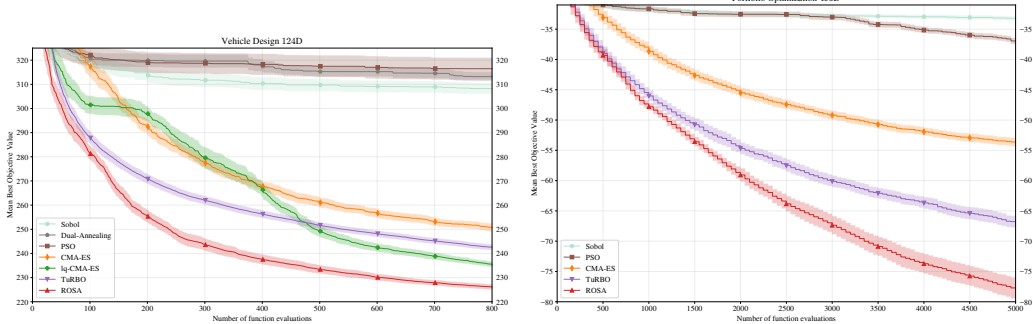

Figure C.7: MOPTA08 optimization in 124 dimensions (left) and Portfolio optimization in 496 dimensions with 50 parallel evaluations (right).

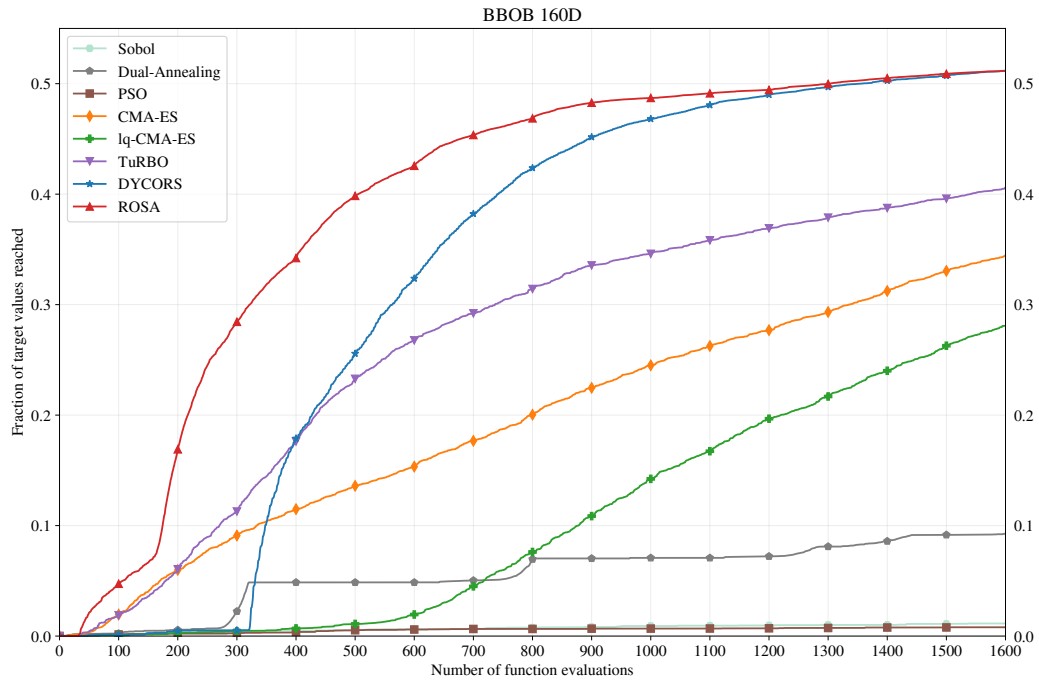

Figure C.8: Empirical cumulative distribution function (for details see [32]) of BBOB optimization in 160 dimensions (higher is better).

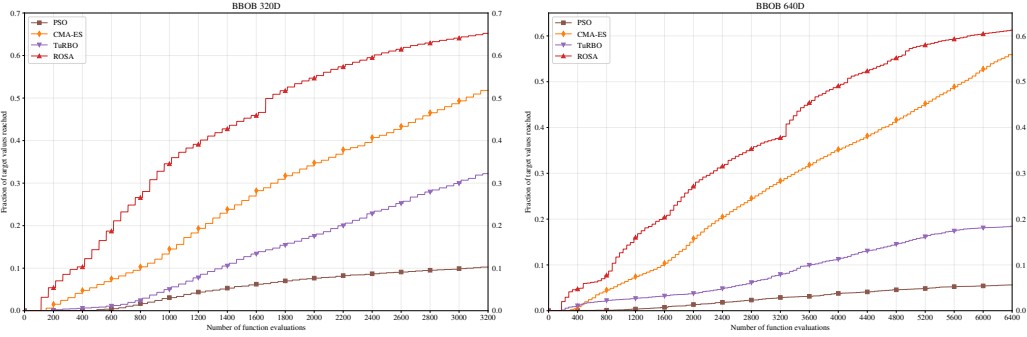

Figure C.9: Empirical cumulative distribution function (for details see [32]) of BBOB optimization in 320 (left) and 640 (right) dimensions with 50 parallel evaluations (higher is better).

## C.1 Wall-clock time comparison

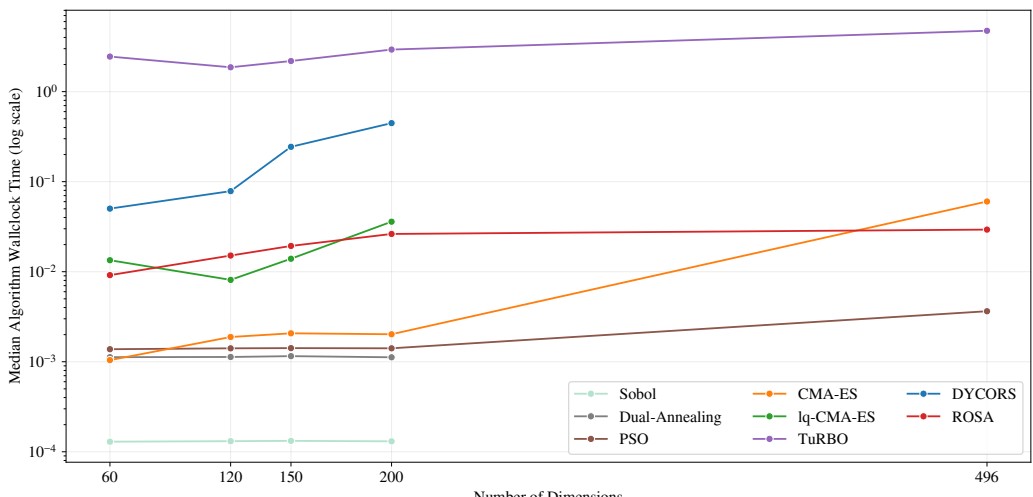

Figure C.10: Wall-clock times of non-evaluation times, i.e. excluding the time it takes to run the function evaluation and only considering the time it takes for an optimization method to propose next evaluation point. Note that the y-axis is in **log scale**.

