# OpenReview forum: "ROSA: An Optimization Algorithm for Multi-Modal Derivative-Free Functions in High Dimensions"
_NeurIPS.cc/2024/Workshop/BDU — NeurIPS BDU Workshop 2024 Poster_

### Official Review · Reviewer_USmT · 2024-09-25

**Rating:** 8
**Confidence:** 2

**Review:**

**Summary**: This work presents a novel optimization algorithm combining stochastic optimization of surrogates with simulated annealing procedures, which offers more computational efficiency than other methods for high-dimensional multi-modal derivative-free optimization problems.

**Evaluation**

- **Quality (8/10)**: The proposal is of evident quality and, in my opinion, meets the scientific standards expected for the workshop.
- **Clarity (8/10)**: Overall, the manuscript conveys its main ideas within the tight limit of four pages. The only point that stands out as unclear to me is the perturbation procedure. As I understand it from Algorithm A.2, noise is sampled from a truncated distribution and then added to the current values along the selected dimensions. How is it ensured that this approach doesn’t push the values outside the bounds of the hypercube? Wouldn’t it be more appropriate for the perturbation to be a sample from a truncated distribution centered around the current value, as in $\hat{x}\sim T(x',\sigma,a,b)$ (as invoked in Theorem 1)?
- **Originality (7/10)**: While the proposal draws heavily on old and established ideas and methods, it innovates by integrating them in a smart modular fashion, obtaining superior performance compared to when these approaches independently feed solutions to the same problem.
- **Significance (8/10)**: Based on its experimental results, I believe the proposed method provides a strong and competitive alternative for optimization in various applied contexts, especially in engineering.

**Pros**
- The method is well-build from the mathematical and methodological point of view.

**Cons**
-  Multi-modal functions are the main focus and are highlighted in the title, yet it appears that both the evaluation and certain theoretical aspects, such as Theorem 1, primarily address the global optimum. It would be helpful to provide more discussion or analysis on local optima as well.
- While there are recommendations regarding the specification of the acceptance probability and perturbation distribution, and the results seem to be robust to variations in these, a more comprehensive ablation study could provide better insights and lead to more informed specifications.

---

### Official Review · Reviewer_Dd5K · 2024-09-26
**Good paper**

**Rating:** 7
**Confidence:** 3

**Review:**

**Summary:**
This manuscript presents a novel optimization algorithm called ROSA, designed for multi-modal, derivative-free optimization in high-dimensional spaces. The algorithm introduces a combination of randomized optimization, simulated annealing, and surrogate optimization, using polyharmonic spline surrogates. ROSA is shown to significantly outperform existing methods on several synthetic benchmark functions and real-world problems while being more computationally efficient.

**Strengths:**
1. The proposed ROSA algorithm addresses a challenging problem in high-dimensional optimization, particularly in scenarios with limited function evaluations.
2. The experimental results demonstrate superior performance across a variety of benchmark functions and outperform competing methods in both accuracy and computational efficiency.
3. The modular design of ROSA allows flexibility in adapting the algorithm to different types of optimization problems, which is a valuable contribution to the field.

**Weaknesses:**
1. The method section lacks clarity regarding how the proposed algorithm is implemented. I think moving the algorithm description from the appendix to the main text would improve readability and accessibility for readers unfamiliar with the approach.
2. The evaluation metric used in the experiments focuses on mean best values across trials. It would also be beneficial to provide visualizations that show how the proposed algorithm and baselines approximate benchmark functions, such as those shown in Figure B.1. This would give readers a clearer understanding of how the algorithm handles different problem landscapes.
3. The figures in the appendix are difficult to read due to the small font size used for labels and numbers. Figures B and C would benefit from larger and more readable captions to ensure that readers can clearly identify the performance differences across methods.

---

### Decision · Program_Chairs · 2024-10-09

Accept (Poster)